# Ameliorated Snake Optimizer-Based Approximate Merging of Disk Wang–Ball Curves

**DOI:** 10.3390/biomimetics9030134

**Published:** 2024-02-22

**Authors:** Jing Lu, Rui Yang, Gang Hu, Abdelazim G. Hussien

**Affiliations:** 1College of Mathematics and Computer Application, Shangluo University, Shangluo 726000, China; 229020@slxy.edu.cn; 2Department of Applied Mathematics, Xi’an University of Technology, Xi’an 710054, China; yangrui2007@163.com; 3Department of Computer and Information Science, Linköping University, 58183 Linköping, Sweden; abdelazim.hussien@liu.se; 4Faculty of Science, Fayoum University, Faiyum 63514, Egypt

**Keywords:** disk Wang–Ball curves, approximate merger, error minimization, modified snake optimizer

## Abstract

A method for the approximate merging of disk Wang–Ball (DWB) curves based on the modified snake optimizer (BEESO) is proposed in this paper to address the problem of difficulties in the merging of DWB curves. By extending the approximate merging problem for traditional curves to disk curves and viewing it as an optimization problem, an approximate merging model is established to minimize the merging error through an error formulation. Considering the complexity of the model built, a BEESO with better convergence accuracy and convergence speed is introduced, which combines the snake optimizer (SO) and three strategies including bi-directional search, evolutionary population dynamics, and elite opposition-based learning. The merging results and merging errors of numerical examples demonstrate that BEESO is effective in solving approximate merging models, and it provides a new method for the compression and transfer of product shape data in Computer-Aided Geometric Design.

## 1. Introduction

Computer-Aided Geometric Design (CAGD for short) [1] takes the representation, drawing, display, analysis, and processing of product geometric shape information as the core research content, and it occupies an important position in manufacturing, medical diagnosis, artificial intelligence, computer vision, and other fields. Product geometry design is the focus of CAGD research, and free-form curves and surfaces are an important tool for describing product geometry. From the Ferguson method [2] to Bézier [3,4], B-spline [5,6], NURBS [7,8,9], and other methods, the representation of free-form curves and surfaces has gone through different stages of development driven by industrial software technology. The Ball method is also one of the current commonly used representations. It was proposed in 1974 as the mathematical basis for the former British Airways CONSURF fuselage surface modeling system [10,11,12]. Subsequent research by several scholars has led to the emergence of a variety of generalized forms such as the Said–Ball curve [13,14,15], the Wang–Ball curve [16], the generalized Ball curves of the Wang–Said type, and the generalized Ball curves of the Said–Bézier type [17]. The Wang–Ball curve not only has good properties such as stability, symmetry, endpoint interpolation, and geometric invariance but also significantly outperforms the Said–Ball and Bézier curves in terms of degree elevation, degree reduction, and recursive evaluation [15].

With the rapid development of the geometric modeling industry, the requirements for accuracy standards, surface quality, and overall smoothness of free curves and surfaces are becoming increasingly stringent. The inaccuracies and limited accuracy caused by the floating-point environment in the modeling of curves and surfaces are major causes for the lack of robustness in solid modeling. Interval analysis [18], which is a tool to enhance the stability of algorithms by dealing with errors, has been gradually introduced into the fields of geometric modeling, computer graphics, and Computer-Aided Design/Computer-Aided Manufacturing (CAD/CAM) since the 1980s [19,20,21]. Subsequently, the concepts of interval Bézier curves, interval Ball curves, and interval B-spline curves have been proposed. Unlike conventional curves, interval curves are constructed with rectangles instead of control vertices represented by real numbers. However, the interval method also has the disadvantage of an expanding error domain under rotational transformation. To this end, Lin et al. [22] put forward disk Bézier curves in combination with the disk algorithm. A disk curve uses disks to represent the control vertices. Compared with interval curves, disk curves have the following advantages: (1) the shape of disk curves remains unchanged under affine transformation; (2) in terms of data storage volume, the interval method requires eight data records in contrast to only two data records for the disk, which reduces the amount of data. Since then, research related to disk curves has been carried out rapidly. Chen [23] studied the order reduction problem of disk Bézier curves using both linear programming and optimization methods. In 2018, Ao et al. [24] proposed a high-precision intersection algorithm for disk B-spline curves and made this curve flexible for stroke representation. Seah et al. [25] applied disk B-spline curves to artistic brush strokes and 2D animation, and Hu et al. [26] constructed the disk Wang–Ball (DWB) curve and investigated its degree reduction problem.

The rapid development of the graphics industry and the manufacturing sector is accompanied by the constant updating of geometric modeling systems, which has led to an increase in the exchange, integration, and sharing of data between different systems for geometric descriptions. Approximate merging [27] is an approximate transformation technique proposed by Hoschek in 1987, which involves approximating a curve consisting of multiple lower-order curve segments with a single higher-order curve. Approximate merging reduces the amount of data transfer during product design and development, thus enabling efficient data transfer and exchange between different systems. In 2001, Hu et al. [28] presented a methodology to merge two adjacent Bézier curves using controlled vertex perturbations and least squares and showed that the merger error could be reduced if the original curves were first raised to higher degrees before approximating the merger. This method is simple, intuitive, and operational, so Tai et al. [29] used a similar method to solve the exact merging of B-spline curves and suggested a node adjustment technique for tuning the end nodes of the *k*th-order curve without varying the shape of the curve. Subsequently, Cheng et al. [30] gave a uniform matrix representation for exact merging by minimizing the curve distance. Zhu and Wang [31] used the *L*_2_ parametrization to measure the approximation error of the curves before and after the merger and achieved the optimal merger problem for Bézier curves with G^2^ continuity. Lu et al. [32] also minimized the *L*_2_ distance and obtained the merger curve for two Bézier curves with G^3^ continuity in an explicit manner.

It is clear from the above research that most of the approximate merging problems are confined to conventional curves and do not involve interval and disk curves. Therefore, the questions of how to extend the approximate merging method to interval as well as disk curves, how to estimate the merging error of interval and circular curves, and whether it is possible to directly extend the approximate merging method of traditional curves to interval curves all need to be addressed. The main objective of this paper is to investigate an approximation method to merge disk Wang–Ball curves with improved robustness and accuracy. The merging error is one of the criteria for judging the effectiveness of curve merging, so we take into account the objective of minimizing the merging error to establish an approximate merging model for circular domain curves. In the problem of how to obtain the optimal coordinates of the merged curves, we invoke meta-heuristic optimization algorithms.

Meta-heuristics algorithms have the advantages of fast convergence, high search power, and high solution accuracy, and they play an increasingly important role in optimization problems. Particle swarm optimization (PSO) [33] has appeared in various improved versions since its introduction in 1995, such as multi-objective particle swarm optimization [34] and adaptive particle swarm optimization algorithms [35], and it has been successfully applied in many fields such as image classification [36], path planning [37] and biomedicine [38]. The marine predators algorithm (MPA) [39] is inspired by ocean predator and prey movements, and its combination with problems such as the shape optimization [40], image segmentation [41], wind power prediction [42], and the 0–1 knapsack problem [43] have achieved high-quality optimization results. There are also a variety of algorithms and enhanced variants developed based on different inspirations, such as the chimp optimization algorithm [44], the nutcracker optimizer [45], enhanced black widow optimization (QIWBWO) [46], the snow ablation optimizer (SAO) [47], the multi-strategy enhanced chameleon swarm algorithm (MCSA) [48], and the multi-objective artificial hummingbird algorithm (MOAHA) [49].

The snake optimizer (SO) [50] is inferred from the unique lifestyle of snakes and has been successfully applied to Hammerstein adaptive filters [51] and power-aware task schedulers for wearable biomedical systems [52]. As SO is prone to falling into local optimality and inadequate optimization capabilities when faced with different optimization problems, enhanced versions of it have been proposed successively to achieve better results. A multi-strategy fused snake optimizer (MFISO) was developed by Fu et al. for deep-learning prediction models of gas prominence in underground mines [53]. Rawa investigated a hybrid form of SO and sine cosine algorithm (SCA) called SO-SCA using both parallel and tandem mechanism runs and used it to solve transmission expansion planning models [54]. Khurma et al. not only generated a binary version called BSO based on the S-shaped transformation function but also integrated the new evolutionary greedy crossover operator with SO to propose a BSO-CV algorithm for medical classification problems [55]. The multi-strategy enhanced snake optimizer (BEESO) [56] is an improved algorithm proposed by Hu et al. for the optimization problem introducing bi-directional search, modified evolutionary population dynamics, and elite opposition-based learning strategy in SO. The experimental results in the literature demonstrate that it possesses a highly competitive search capability and convergence speed compared to a variety of sophisticated algorithms. Therefore, BEESO will be used to solve the approximate merging problem for DWB curves, and the main contributions are summarized as follows:(1)We discuss the approximate merging problem of DWB curves and establish an approximate merging optimization model with the merging error as the objective;(2)We propose an approximate merging method of DWB curves based on BEESO and demonstrate the optimization capability of BEESO with numerical examples.

The remainder of this text is structured as follows: The definition of DWB curves is presented in Section 2, along with a discussion of the approximate merging of adjacent DWB curves and a specific optimization model; Section 3 introduces the BEESO and proposes an approximate merging method of DWB curves based on BEESO and gives three numerical examples; Section 4 concludes the paper.

## 2. Approximate Merging of DWB Curves

### 2.1. Definition of the DWB Curves

**Definition** **1.**
*In R^2^, given n+1 control disks, then the DWB curve of degree n is defined as*

(1)
$$(W)(t)=\sum_{i=0}^{n}W_{i,n}(t)(P_{i})=\sum_{i=0}^{n}W_{i,n}(t)(p_{i},r_{i}),(0\leq t\leq 1),$$

*where $p_{i}=(x_{i},y_{i})$ and r_i_ represent the center coordinates and radius of the control disk, and ${\left\{W_{i,n}(t)\right\}}_{i=0}^{n}$ indicates Wang–Ball basis functions, in which*

(2)
$$W_{i,n}(t)=\left\{\begin{array}{ll} {\left(2t\right)}^{i}{\left(1-t\right)}^{i+2}, & 0\leq i\leq \left\lfloor n/2\right\rfloor -1,\;\\ {\left(2t\right)}^{\left\lfloor n/2\right\rfloor }{\left(1-t\right)}^{\left\lceil n/2\right\rceil }, & i=\left\lfloor n/2\right\rfloor ,\;\\ {\left(2(1-t)\right)}^{\left\lfloor n/2\right\rfloor }t^{\left\lceil n/2\right\rceil }, & i=\left\lceil n/2\right\rceil ,\;\\ W_{n-i}^{n}(1-t), & \left\lceil n/2\right\rceil +1\leq i\leq n.\; \end{array}\right.$$


*According to Equation (1), the DWB curve can be written in the following form:*

(3)
$$(W)(t)=(C(t),R(t))=(\sum_{i=0}^{n}W_{i}^{n}(t)p_{i},\sum_{i=0}^{n}W_{i}^{n}(t)r_{i}),(0\leq t\leq 1),$$

*where **C**(t) and R(t) are the center curve and radius function, respectively.*


### 2.2. Approximate Merging of Adjacent DWB Curves

#### 2.2.1. Problem Description

It is given that ${\left(W\right)}_{1}(t)$ and ${\left(W\right)}_{2}(t)$ are two adjacent DWB curves, and their control disks are $\left(P_{1,i})(i=0,1,\cdots ,n_{1}\right)$ and $\left(P_{2,j})(j=0,1,\cdots ,n_{2}\right)$, respectively. The approximate merging of these two neighboring DWB curves means seeking another *n*th DWB curve (D)(t), such that the metric distance between (D)(t) and $\left(\bar{D})(t\right)$ is minimized on the interval [0, 1]. Here, the expression is
(4)$$(\bar{D})(t)=\left\{\begin{array}{l} \sum_{i=0}^{n_{1}}W_{i,n_{1}}(\frac{t}{\lambda })(P_{1,i}),(0\leq t\leq \lambda ),\\ \sum_{j=0}^{n_{2}}W_{j,n_{2}}(\frac{t-\lambda }{1-\lambda })(P_{2,j}),(\lambda \leq t\leq 1), \end{array}\right.$$
where *λ* is the subdivision parameter. $W_{i,n_{1}}(\frac{t}{\lambda })$ and $W_{j,n_{2}}(\frac{t-\lambda }{1-\lambda })$ are the Wang–Ball basis functions of the order *n*_1_ and *n*_2_ defined by Equation (2). $d((D)(t),(\bar{D})(t))$ can be selected to the appropriate value as required.

#### 2.2.2. Construction of the Approximate Merger Model

Approximate merging without endpoint preservation

According to Equation (3), the two adjacent DWB curves ${\left(W\right)}_{1}(t)$, ${\left(W\right)}_{2}(t)$ are denoted as $\left(C_{1}(t),R_{1}(t)\right)$ and $\left(C_{2}(t),R_{2}(t)\right)$, and the DWB curve $\left(\bar{D})(t\right)$ in Equation (4) is expressed as $\left(\bar{C}(t),\bar{R}(t)\right)$. Referring to the *n*th DWB curve (D)(t) to be found as the third curve, that is,
(5)$$(D)(t)=(C(t),R(t))=\sum_{k=0}^{n}W_{k,n}(t)(Q_{k}),n\geq \max(n_{1},n_{2}).$$

Take the subdivision parameter *λ* to be any constant in the open interval (0, 1). Then, divide (D)(t) into two DWB curves of degree *n*_1_ on the left and *n*_2_ on the right, which are recorded as ${\left(D\right)}_{left}(t)=(C_{left}(t),R_{left}(t))$ and ${\left(D\right)}_{right}(t)=(C_{right}(t),R_{right}(t))$,
(6)$$\left\{\begin{array}{l} {\left(D\right)}_{left}(t)=(D)(\lambda t),\\ {\left(D\right)}_{right}(t)=(D)(\lambda +(1-\lambda )t), \end{array}\right.t\in [0,1].$$

For the three curves above, we measure their metric distances in terms of two components, the center curve and the radius function. Let the subdivision parameter $\lambda =\frac{\int_{0}^{1}|{C^{\prime }}_{1}(t)|dt}{\int_{0}^{1}|{C^{\prime }}_{1}(t)|dt+\int_{0}^{1}|{C^{\prime }}_{2}(t)|dt}$, the distance between the center curves, which is defined as
(7)$$d_{C}(C(t),\bar{C}(t))=\int_{0}^{1}|C_{1}(t)-C_{left}(t){|}^{2}dt+\int_{0}^{1}|C_{2}(t)-C_{right}(t){|}^{2}dt.$$

Based on the above description of curve merging, there should be
(8)$$d_{C}(C(t),\bar{C}(t))=\min.$$

For the same between the radius functions R(t) and $\bar{R}(t)$, there is
(9)$$\min\;d_{R}(R(t),\bar{R}(t))=\int_{0}^{1}|R_{1}(t)-R_{left}(t){|}^{2}dt+\int_{0}^{1}|R_{2}(t)-R_{right}(t){|}^{2}dt.$$

In order to verify the optimized effect, the merging error formula is a common criterion for judging. In summary, this paper will use the merging error between the merging curve ${\left(W\right)}_{1}(t)$ and ${\left(W\right)}_{2}(t)$ as the curve to be merged and as an objective function to build an optimization model, defined as
(10)$$\min\;\varepsilon =\int_{0}^{1}\left(|C_{1}(t)-C_{left}(t){|}^{2}+|C_{2}(t)-C_{right}(t){|}^{2}\right)dt+\int_{0}^{1}\left(|R_{1}(t)-R_{left}(t){|}^{2}+|R_{2}(t)-R_{right}(t){|}^{2}\right)dt.$$

2.Approximate merging with endpoint preservation

The resulting merged curve does not guarantee that (D)(t) is interpolated at the left endpoint of the curve ${\left(W\right)}_{1}(t)$ and the right endpoint of the curve ${\left(W\right)}_{2}(t)$, so this part will build an approximate merge optimization model with endpoint-preserving interpolation between the merged curve and the curve to be merged. Endpoint-preserving interpolation means that the third curve (D)(t) found must not only approximate the merge of the two given DWB curves, but also the interpolation at the left endpoint of ${\left(W\right)}_{1}(t)$ as well as the right endpoint of ${\left(W\right)}_{2}(t)$ to be merged.

In accordance with the above definition of endpoint-preserving interpolation approximation merging and the endpoint properties of DWB curves, the endpoint-preserving interpolation approximation merging optimization model is expressed as
(11)$$\min\;\varepsilon =\int_{0}^{1}\left(|C_{1}(t)-C_{left}(t){|}^{2}+|C_{2}(t)-C_{right}(t){|}^{2}\right)dt+\int_{0}^{1}\left(|R_{1}(t)-R_{left}(t){|}^{2}+|R_{2}(t)-R_{right}(t){|}^{2}\right)dt.$$

The constraints are as follows:(12)$$\left\{\begin{array}{l} (Q_{0})=(q_{0},r_{0})=(p_{1,0},r_{1,0})=(P_{1,0}),\\ (Q_{n})=(q_{n},r_{n})=(p_{2,n_{2}},r_{2,n_{2}})=(P_{2,n_{2}}). \end{array}\right.$$

## 3. DWB Curves Merging Based on BEESO

### 3.1. BEESO

BEESO [56] is an improved algorithm proposed to address the shortcomings of SO, which integrates three strategies into SO to improve its optimization capabilities. The update phase of BEESO includes exploration, exploitation, and mutation operations. The initial population is calculated using Equation (13), in which $y_{i}=[y_{i,1},y_{i,2},\cdots ,y_{i,D}](i=1,2,\cdots ,N)$ represents the *i*th individual in a population.
(13)$$y_{i}=LB+r\times (UB-LB),$$
where *r* is randomly selected at [0, 1], ***LB*** stands for the lower bound, and ***UB*** stands for the upper bound.

The population is composed of 50% females and 50% males, and the population is randomly split into two sub-groups before the start of the iteration, as follows:(14)$$N_{m}\approx N/2,\;N_{f}=N-N_{m}$$
where *N_m_* and *N_f_* represent male and female subgroups.

The food mass *P* and temperature *Temp* that control the algorithmic process are calculated with the following formula:(15)$$Temp=\exp(\frac{-k}{K}),$$
(16)$$P=s_{1}\times \exp(\frac{k-K}{K}),$$
where *k* and *K,* respectively, represent the current and maximum number of iterations, and *s*_1_ = 0.5.

#### 3.1.1. Exploration Phase

If the food quality *P* < *Threshold* means that it is of low quality, the population needs to search for better-quality food in the feasible range, which indicates that BEESO enters the exploration phase. The exploration phase consists of a random search and a bi-directional search; BEESO will use these two search methods to update the position of individuals and select the better ones for the next stage by comparing their fitness values. For the random search, the random update position of each individual is formulated as follows:(17)$$y_{i,m}(k+1)=y_{r,m}(k)\pm s_{2}\times AM\times ((UB-LB)\times r+LB),$$
(18)$$y_{i,f}(k+1)=y_{r,f}(k)\pm s_{2}\times AF\times ((UB-LB)\times r+LB),$$
where $y_{r,m}$ and $y_{r,f}$ represent male and female individuals, respectively. *AM* and *AF* are the snake’s ability to find food, *r* is a random number, and *s*_2_ = 0.5.
(19)$$y_{i,m}(k+1)=y_{r,m}(k)\pm s_{2}\times AM\times ((UB-LB)\times r+LB),$$
(20)$$y_{i,f}(k+1)=y_{r,f}(k)\pm s_{2}\times AF\times ((UB-LB)\times r+LB),$$
where $Fit_{•}$ represents the fitness value.

Bi-directional search makes use of the best and worst individuals to guide BEESO toward the optimal value while maximizing the search area, which effectively improves the disadvantages of the random search such as lower randomness, higher uncertainty, and a narrow search range. The mathematical approach is formulated as
(21)$$y_{i,m}(k+1)=y_{i,m}(k)+r_{1}\times (y_{best,m}-y_{i,m}(k))-r_{2}\times (y_{worst,m}-y_{i,m}(k))$$
(22)$$y_{i,f}(k+1)=y_{i,f}(k)+r_{1}\times (y_{best,f}-y_{i,f}(k))-r_{2}\times (y_{worst,f}-y_{i,f}(k))$$
where $y_{best}$ and $y_{worst}$ are the best and worst individuals, respectively, and *r*_1_ and *r*_2_ are evenly generated random numbers.

#### 3.1.2. Exploitation Phase

A higher quality of food represents BEESO entering the exploitation phase. When the temperature *Temp* > 0.6 represents a higher temperature in the environment, the snake will move toward the food, as described by the mathematical equation
(23)$$y_{i,j}(k+1)=y_{food}\pm c_{3}\times Temp\times r\times (y_{food}-x_{i,j}(k))$$

Mating behavior occurs when the temperature is right, and there is a fighting mode and a mating mode. Fighting mode means that each female engages in mating with the best male, and the males will get the best females by fighting, as in the following equation:(24)$$y_{i,m}(k+1)=y_{i,m}(k)\pm s_{3}\times FM\times r\times (y_{best,f}-y_{i,m}(k))$$
(25)$$y_{i,f}(k+1)=y_{i,f}(k)\pm s_{3}\times FF\times r\times (y_{best,m}-y_{i,f}(k))$$
where *FM* and *FF* are the combat abilities and are calculated using the following formula:(26)$$FM=\exp(\frac{-Fit_{best,f}}{Fit_{i}})$$
(27)$$FF=\exp(\frac{-Fit_{best,m}}{Fit_{i}})$$

Mating patterns in which mating between each pair of individuals occurs is mathematically modeled as
(28)$$y_{i,m}(k+1)=y_{i,m}(k)\pm s_{3}\times MM\times r\times (P\times y_{i,f}(k)-y_{i,m}(k))$$
(29)$$y_{i,f}(k+1)=y_{i,f}(k)\pm s_{3}\times MF\times r\times (P\times y_{i,m}(k)-y_{i,f}(k))$$
where *MM* and *MF* stand for the mating ability:(30)$$MM=\exp(\frac{-Fit_{i,f}}{Fit_{i,m}})$$
(31)$$MF=\exp(\frac{-Fit_{i,m}}{Fit_{i,f}})$$

There is a potential for egg production after mating. If the snake eggs hatch, modified evolutionary population dynamics (MEPD) is implemented on the current parent to improve population quality by eliminating the poorer individuals and mutating the better ones. New offspring are first generated for the bottom 50% of individuals using Equations (32) and (33):(32)$$Offspring_{i}(k+1)=\left\{\begin{array}{l} y_{best}(k)+sign(r-0.5)\times (UB-LB\times r+LB),\;\mathrm{if}\;r<0.5\\ y_{i}(k)+sign(r-0.5)\times (UB-LB\times r+LB),\;\mathrm{else} \end{array}\right.$$
(33)$$y_{i}(k+1)=\left\{\begin{array}{l} Offspring_{i}(k+1),\mathrm{if}\;Fit(Offspring_{i}(k+1))<Fit(y_{i}(k+1))\\ UB-LB\times r+LB,\;\mathrm{else} \end{array}\right.$$
where *i* = 1, 2, ..., *N*/2.

The mutation operation is applied to the top 50% of individuals, as follows:(34)Myi(k)=y(k)p1+F⋅(y(k)p2−y(k)p3)
(35)$$y_{i}(k+1)=\left\{\begin{array}{l} My_{i}(k),\mathrm{if}\;Fit(My_{i}(k))<Fit(y_{i}(k+1))\\ y_{i}(k+1),\;\mathrm{else} \end{array}\right.$$
where *p*_1_, *p*_2_, *p*_3_ are random integers between [1, *N*], and *p*_1_ ≠ *p*_2_ ≠ *p*_3_ ≠ *i*. *F* is the scaling factor:(36)$$F=\frac{1}{2}(\sin(2\pi \times freq\times k)\times (k+K)+1)$$
where *freq* is the frequency of vibration of the sine function.

#### 3.1.3. Elite Opposition-Based Learning Strategy

The tendency to fall into local optima is a common problem with optimization algorithms. The elite opposition-based learning strategy is meant to optimize the better-performing individuals in the population, so that the algorithm approximates the global optimum with a higher probability. $Ey_{n}=[ey_{n,1},ey_{n,2},\cdots ,ey_{n,D}],(n=1,2,\cdots ,EN)$ stands for elite individuals, who rank in the top *EN* of the population. The elite opposite solution of the current individual is calculated below:(37)$$e{\bar{y}}_{i,j}(k)=S\cdot (EA_{j}(k)+EB_{j}(k))-y_{i,j}(k)$$
(38)$$EB_{j}(k)=\max(ey_{n,j}(k))$$
(39)$$e{\bar{y}}_{i,j}(k)=rand\cdot (EB_{j}(k)-EA_{j}(k))+EA_{j}(k),if\;e{\bar{y}}_{i,j}<LB_{j}||e{\bar{y}}_{i,j}>UB_{j}$$
where *EN* = 0.1 × *N*, and EA(k) and EB(k) are the minimum and maximum values of elite individuals.

### 3.2. Steps for Solving the Approximate Merger Models by BEESO

In Section 2, two optimization models, endpoint-preserving merging and non-endpoint-preserving merging, are established with the merging error of the curves before and after merging as the objective function. In this section, the BEESO will be used to solve the approximate merged optimized model for DWB curves, and the implementation steps are as follows:

Step one: Setting the parameters of the algorithm;

Step two: Enter the coordinates and radius of the DWB curves to be merged, and calculate the subdivision parameters;

Step three: Initialization. Calculate the initial population according to Equation (13), and divide it into female and male populations. Use the approximate merging model Equation (9) (or Equation (10)) as the objective function;

Step four: Judging food quality. If *P* < 0.25, execute Equations (17)–(20) and the bi-directional search Equations (21) and (22) of the exploration phase to generate new individuals. Calculate the fitness value of the individuals, and select the better individuals to proceed to the next phase; otherwise, proceed to the exploitation phase;

Step five: The exploitation phase starts by calculating the temperature *Temp*. If *Temp* > 0.6, the individual moves closer to the food as in Equation (23). Otherwise, it enters the fight mode or the mating mode. When *r* > 0.6, the individual is in fight mode and performs Equations (24)–(27); otherwise, it operates in mating mode;

Step six: The mating pattern performs Equations (28)–(31) on individuals, followed by consideration of whether the eggs hatch after mating. If the eggs hatch, MEPD (Equations (32)–(36)) is used to produce individuals that move on to the next stage;

Step seven: Find the elite individuals, and update the population again by Equations (37)–(39);

Step eight: Calculate the fitness value, and determine the optimal individual;

Step nine: Determine the termination condition of the algorithm. Let *k* = *k* + 1. If k<K, return to step four; otherwise, output the minimum merging error and the coordinates of the control disks of the merging curve.

A detailed flowchart for the approximate merged model is shown in Figure 1. 

### 3.3. Optimization Examples

To verify the feasibility of the algorithm, three specific numerical examples are given in this section. Each numerical example is optimized in terms of both endpoint-preserving interpolation and endpoint-unperceiving interpolation. The population size and iterations in all experiments are 50 and 500. In addition to BEESO, several advanced algorithms are selected for comparison purposes, such as CSA [48], SCA [57], grey wolf optimizer (GWO) [58], white shark optimizer (WSO) [59], the Coot optimization algorithm (COOT) [60], the sooty tern optimization algorithm (STOA) [61], and Harris hawks optimization (HHO) [62]. 

**Example** **1.***It is given that two adjacent DWB curves and their control vertex coordinates as well as the control radius are shown in Table 1, and this constructs the shape of the DWB curve shown in Figure 2. In this example, these two curves of 3 and 4 degrees will be combined into a single 6-degree DWB curve according to the model built in Section 2.2.2*.

Table 2 and Table 3 show the optimum results obtained by the eight intelligent algorithms including BEESO and SCA for the two cases of no endpoint preservation and endpoint preservation, respectively, which include the control disk coordinates of the merged curves and the merging error. The comparative results of the before and after merging curves are shown in Figure 3 and Figure 4, where the merged DWB curve is shown in green. The change in the shape of the curves before and after the merger can be observed in the figures. For example, the shape of the left segment of the curve obtained by CSA shows a large error. In addition, the convergence in the optimization process is shown in Figure 3i and Figure 4i. 

The overall result shows that the optimization of the “C” curve fits the original curve better. For the non-endpoint-preserving case, SCA and STOA obtain curves that differ significantly from the width of the original curve, and their errors are accordingly two of the larger of all the algorithms. In the endpoint-preserving case, STOA also has a larger difference in results and a slower convergence rate in the optimization process. Although the remaining algorithms have similar results, BEESO has better results when looking at the fit and width of the two curves before and after merging. BEESO achieves the best results among the eight algorithms, both in terms of error results and convergence speed.

**Example** **2.**
*The purpose of this example is to approximate the merging of two adjacent 4-degree DWB curves into one 5-degree curve. The control disks of the two curves before merging are shown in Table 4, and the shape of the curve obtained by visualizing it is shown in Figure 5.*


Table 5 and Table 6, respectively, give the optimization results of the eight algorithms for the two cases of no endpoint preservation and endpoint preservation, which include the optimal control vertices and the merging error. Figure 6 and Figure 7 show the optimization results for two adjacent 4-degree DWB curves merged into one 5-degree DWB curve under different constraints, respectively. 

In the case where endpoints are not preserved, there is a large difference in the optimization results between the eight algorithms. For example, the visual difference between the merged curves obtained by HHO, GWO, and CSA and the original curve is significant, which is also evident from the error for each algorithm in Table 5. Whereas for the endpoint-preserving case, BEESO, CSA, SO, WSO, and COOT obtain similar effect plots, the optimized curve from GWO has a large difference in width from the pre-merged curve. However, by combining the error data in the table, BEESO obtains the minimum error under each of the two constraints.

**Example** **3.**
*A 3-degree curve and 4-degree curve are given, respectively, as shown in Table 7, and the two curves are constructed as shown in Figure 8. This example will discuss the problem of merging 3- and 4-degree curves based on intelligent algorithms in both the non-endpoint-preserving and endpoint-preserving cases.*


The results obtained for this example using the eight algorithms including BEESO and CSA are shown in Table 8 and Table 9. In addition, the graphs of the merging effect obtained by visualizing the control disks are shown in Figure 9 and Figure 10. 

For the case of non-endpoint preservation, the approximate merging results of the three algorithms BEESO, COOT, and SO in Figure 9 are relatively good, with BEESO having the smallest merging error of all the algorithms. The other algorithms, on the other hand, all have much room for improvement in their overall results, especially GWO, HHO, and STOA, which have large errors in the position of the curve and the width of the curve before merging. Due to the low dimensionality of the optimizing variables in the endpoint-preserving case, the results for all seven algorithms apart from STOA are approximate and have a small gap in terms of the combined errors given. For BEESO, SO, and WSO, there is no significant gap when the errors are kept to three decimal places. The experimental outcome by keeping the valid data to more than one decimal place is 0.14669880, 0.14669887, and 0.14669885 for BEESO, SO, and WSO respectively; the error for BEESO is the smallest of the three algorithms.

## 4. Conclusions

Based on the basic theory of the disk Wang–Ball curve, an approximate merging method based on a meta-heuristics algorithm is proposed for the problem that this curve is difficult to merge. This paper first discusses the approximate merging of DWB curves and establishes two optimization models from the perspectives of non-endpoint-preserving merging and endpoint-preserving merging, with the merging error as the objective function. In addition, BEESO is introduced to solve the constructed optimization models, and the steps of the algorithm for solving the approximate merging problem of DWB curves are specified. The method can directly obtain the control disks of the merging curve while calculating the merging error, which is characterized by simple and practical calculation. Finally, some advanced algorithms are selected for comparison in Section 3.3, and the effectiveness of the method in curve design is also demonstrated by the fact that BEESO achieves DWB curves with good merging results in all three numerical examples.

The approximate merging model developed in this paper achieves a good merging effect and a small error, but the integral form leads to a more complex model, and the algorithm runs slower during the solution process. The question of how to simplify or build the mathematical model in a simple way so that it can be applied to the approximate merging of disk curves is a problem to be solved in the future. In addition, BEESO can be used to solve optimization problems in areas such as cryptosystem design [63], path planning, geometry optimization [40,64], engineering design [65,66], and feature selection [67].

## Figures and Tables

**Figure 1 biomimetics-09-00134-f001:**
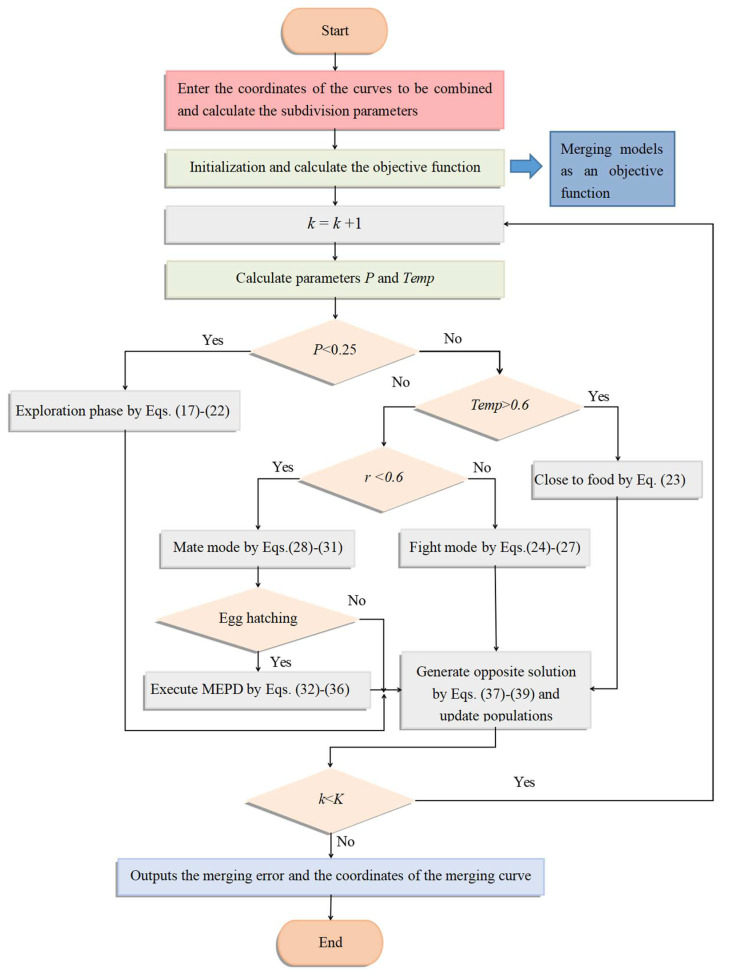
Flowchart of BEESO solving the approximate merged model.

**Figure 2 biomimetics-09-00134-f002:**
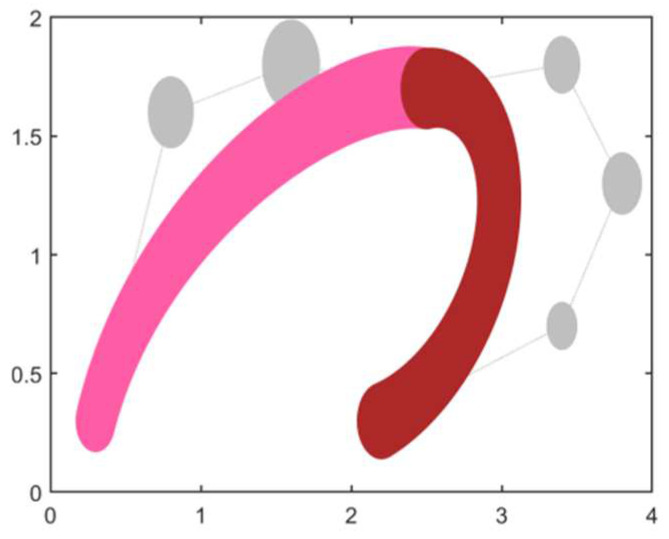
Adjacent 3-degree and 4-degree DWB curves in Example 1.

**Figure 3 biomimetics-09-00134-f003:**
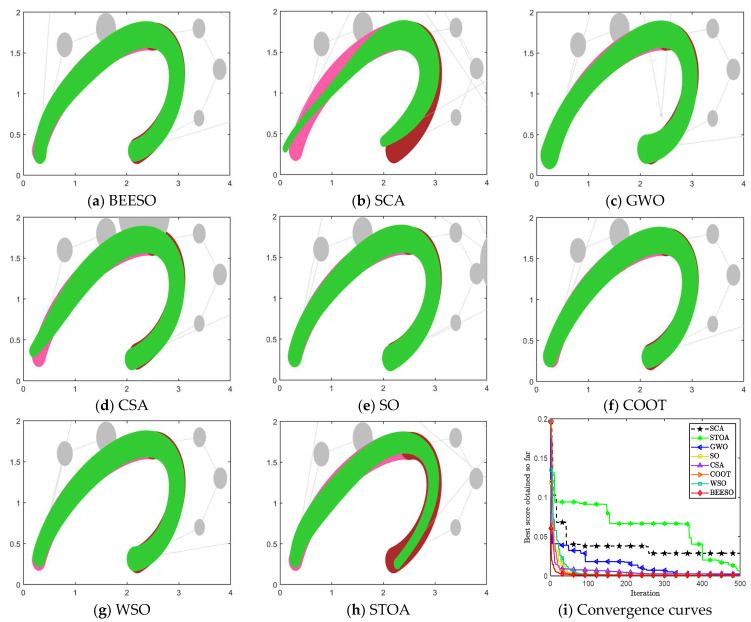
Graphs and convergence curves for merging into 6-degree curves without endpoint preservation in Example 1.

**Figure 4 biomimetics-09-00134-f004:**
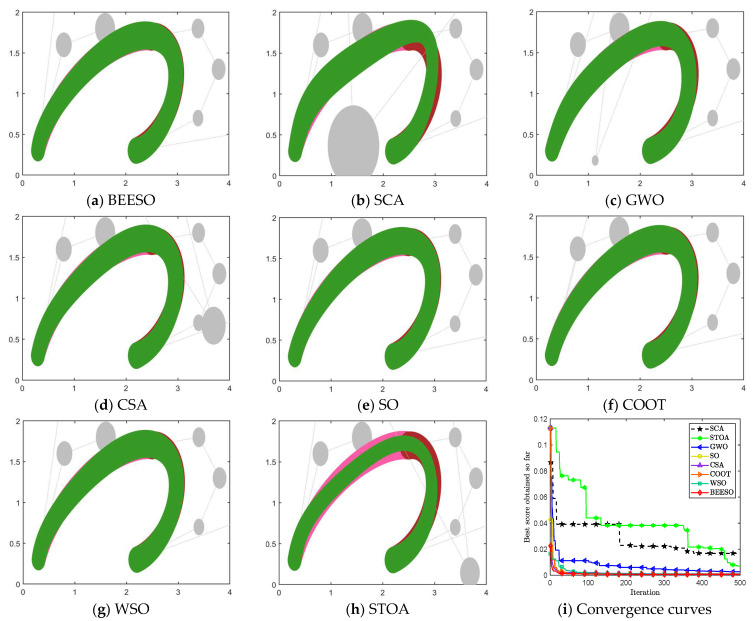
Graphs and convergence curves for merging into 6-degree curves in the endpoint-preserving case.

**Figure 5 biomimetics-09-00134-f005:**
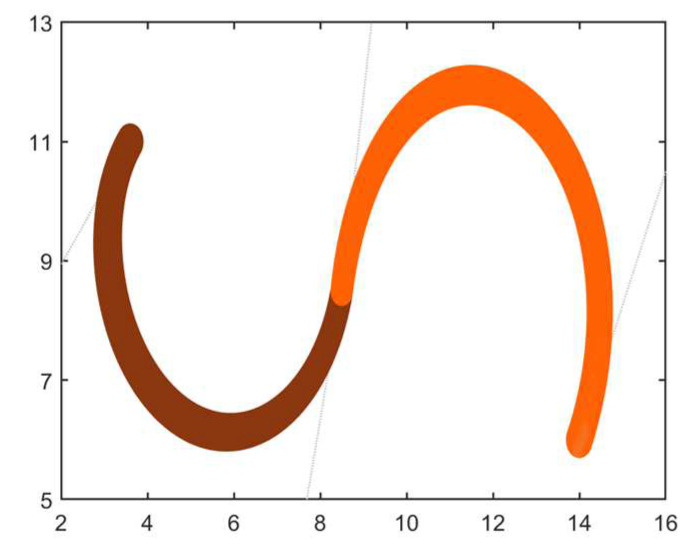
Two adjacent 4-degree DWB curves in Example 2.

**Figure 6 biomimetics-09-00134-f006:**
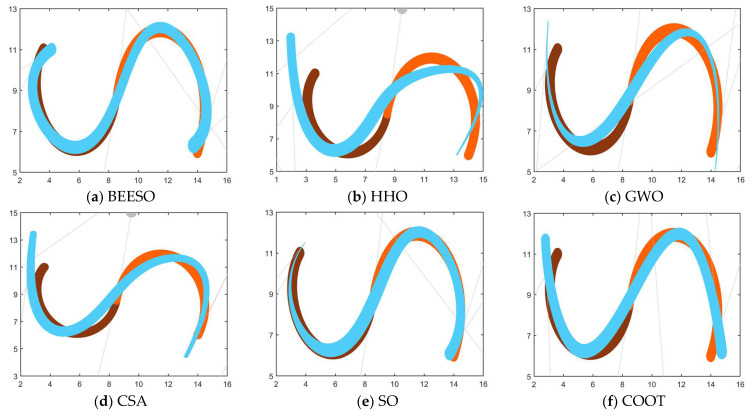
Graphs and convergence curves for merging into 6-degree curves without endpoint preservation in Example 2.

**Figure 7 biomimetics-09-00134-f007:**
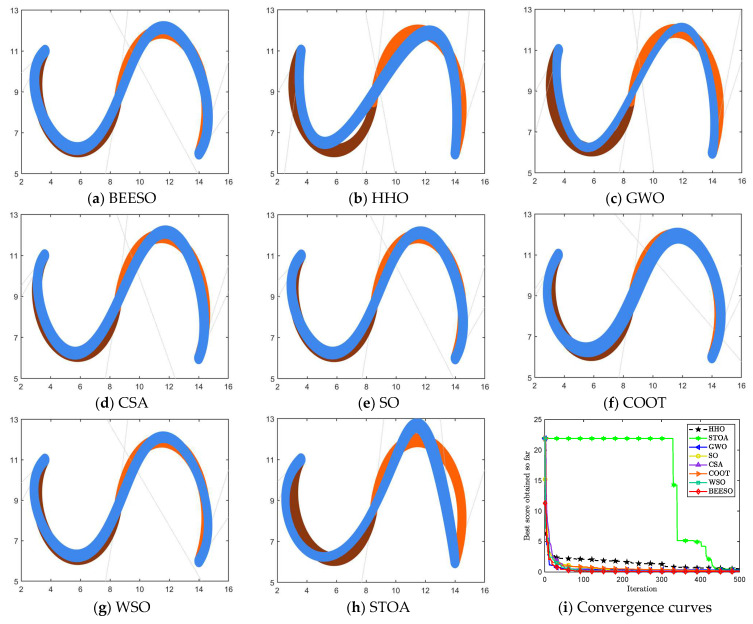
Graphs and convergence curves for merging into 5-degree curves in the endpoint-preserving case.

**Figure 8 biomimetics-09-00134-f008:**
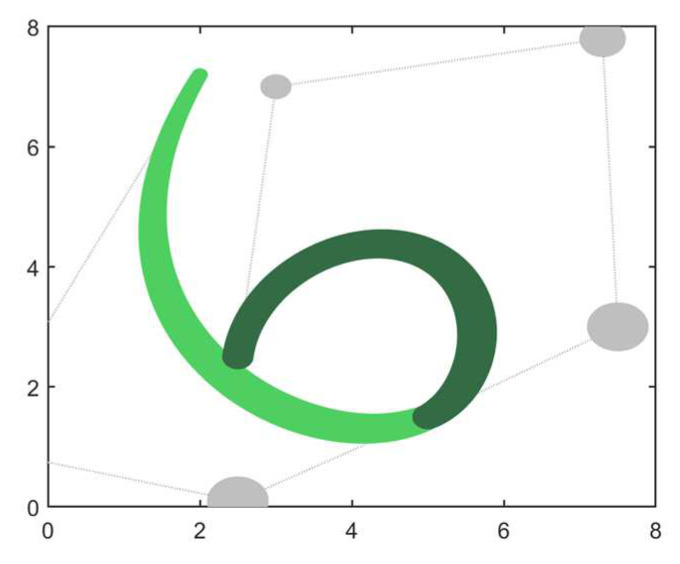
Adjacent 3-degree and 4-degree DWB curves in Example 3.

**Figure 9 biomimetics-09-00134-f009:**
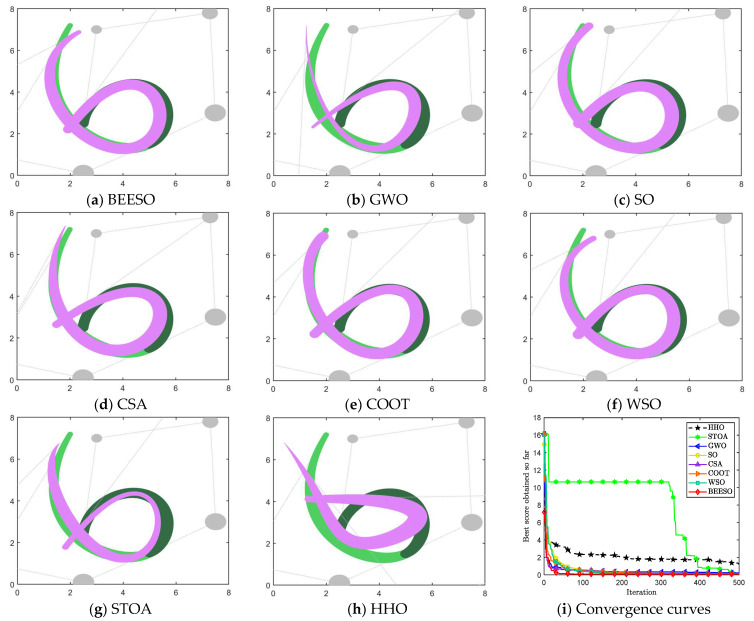
Graphs and convergence curves for merging into 4-degree curves without endpoint preservation.

**Figure 10 biomimetics-09-00134-f010:**
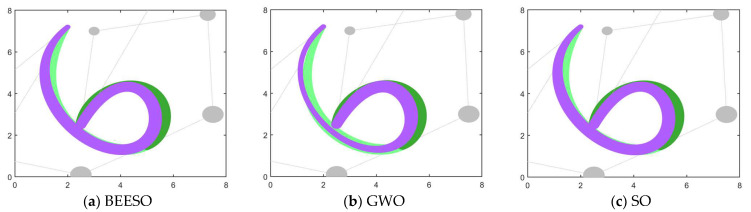
Graphs and convergence curves for merging into 4-degree curves in the endpoint-preserving case.

**Table 1 biomimetics-09-00134-t001:** Coordinates of the two adjacent DWB curves in Example 1.

Curves	Variables	Values				
		*i* = 0	*i* = 1	*i* = 2	*i* = 3	*i* = 4
(W)1(t)	** *p* ** _1,*i*_	(0.3, 0.3)	(0.8, 1.6)	(1.6, 1.8)	(2.5, 1.7)	/
	*r* _1,*i*_	0.13	0.15	0.19	0.17	/
${\left(W\right)}_{2}(t)$	** *p* ** _2,*i*_	(2.5, 1.7)	(3.4, 1.8)	(3.8, 1.3)	(3.4, 0.7)	(2.2, 0.3)
	*r* _2,*i*_	0.17	0.12	0.13	0.1	0.16

**Table 2 biomimetics-09-00134-t002:** Optimization results of merging into 6-degree DWB curves without endpoint preservation.

Methods	Variables	Optimal Results							Errors
		*j* = 0	*j* = 1	*j* = 2	*j* = 3	*j* = 4	*j* = 5	*j* = 6	
BEESO	*x_j_*	0.32061	0.64240	5.77004	1.14708	5.71427	5.46150	2.16106	7.256 × 10^−4^
	*y_j_*	0.26396	3.21220	1.23437	3.87470	7.90834	0.93425	0.30057	
	*r_j_*	0.12286	0.14662	0.59951	0.06383	0.19417	0.19542	0.13245	
SCA	*x_j_*	0.10000	1.95487	4.71265	0.50000	6.16513	5.45862	2	2.884 × 10^−2^
	*y_j_*	0.32289	2.79650	0.17982	5.16130	4.69827	1.69757	0.40675	
	*r_j_*	0.05000	0.01920	0.07688	0.48184	0.00154	0.60000	0.06559	
GWO	*x_j_*	0.24801	1.67657	2.40646	2.88812	2.41048	6.34649	2.12232	1.450 × 10^−3^
	*y_j_*	0.25249	3.38514	0.73713	3.98825	8.35052	0.67943	0.32403	
	*r_j_*	0.17159	0.01329	0.01217	0.58550	0.02833	0.01154	0.17778	
CSA	*x_j_*	0.19345	2.33305	1.02304	3.01595	1.85392	6.61869	2.09672	2.713 × 10^−3^
	*y_j_*	0.36494	2.07844	3.39108	4.04875	6.32216	1.55162	0.26369	
	*r_j_*	0.07595	0.49882	0.02186	0.15704	0.03753	0.23745	0.13200	
SO	*x_j_*	0.29201	1.10659	4.26885	1.84796	4.38502	5.84374	2.13823	1.009 × 10^−3^
	*y_j_*	0.29032	2.95108	1.44851	4.31903	6.39291	1.45585	0.26852	
	*r_j_*	0.13274	0.08649	0.39728	0.39219	0.04970	0.09040	0.15708	
COOT	*x_j_*	0.26103	1.51427	3.24220	2.07129	4.64899	5.61018	2.16634	9.550 × 10^−4^
	*y_j_*	0.30621	2.66449	2.57976	3.60219	7.88557	1.01002	0.29692	
	*r_j_*	0.13240	0.21687	0.11664	0.27394	0.01746	0.22524	0.13620	
WSO	*x_j_*	0.30302	0.99589	4.34925	2.04026	4.08624	5.85888	2.14905	8.701 × 10^−4^
	*y_j_*	0.33993	2.15657	4.13252	2.94350	9.02235	0.70568	0.31241	
	*r_j_*	0.12336	0.14994	0.56790	0.02366	0.55272	0.02905	0.15067	
STOA	*x_j_*	0.29616	0.97048	5.28889	0.87130	8.45277	4.23581	2.27127	5.721 × 10^−3^
	*y_j_*	0.31909	2.91011	0.60983	5.06904	5.36542	1.73980	0.25750	
	*r_j_*	0.10937	0.42115	0.01206	0.00742	0.57521	0.00319	0.06925	

**Table 3 biomimetics-09-00134-t003:** Optimization results of merging into 6-degree DWB curves in the endpoint-preserving case.

Methods	Variables	Optimal Results					Errors
		*j* = 1	*j* = 2	*j* = 3	*j* = 4	*j* = 5	
BEESO	*x_j_*	0.71364	6.03125	0.74095	6.90557	5.00301	5.777 × 10^−4^
	*y_j_*	2.49972	3.72707	2.64980	9.64966	0.6	
	*r_j_*	0.19303	0.06100	0.48497	0.14903	0.01804	
SCA	*x_j_*	1.23993	1.42726	5.46332	2.16787	5.65829	1.682 × 10^−2^
	*y_j_*	3.53095	0.36503	3.70241	8.47249	1.11223	
	*r_j_*	0.36242	0.49446	0.00100	0.00208	0.00223	
GWO	*x_j_*	1.73337	1.13356	3.80688	2.03884	6.01318	2.569 × 10^−3^
	*y_j_*	3.27833	0.18116	4.62612	6.98556	1.09384	
	*r_j_*	0.34868	0.06189	0.10914	0.22476	0.06069	
CSA	*x_j_*	1.20266	3.68944	2.18905	4.66729	5.46518	1.108 × 10^−3^
	*y_j_*	3.16217	0.66512	4.41674	7.22810	1.04698	
	*r_j_*	0.18345	0.22954	0.30863	0.32153	0.00197	
SO	*x_j_*	0.97965	4.85955	1.39269	5.93192	5.20424	6.454 × 10^−4^
	*y_j_*	2.44741	3.73946	2.83659	9.20922	0.70607	
	*r_j_*	0.14520	0.33054	0.28144	0.34953	0.00432	
COOT	*x_j_*	1.28107	3.44674	2.22999	4.73317	5.43497	8.813 × 10^−4^
	*y_j_*	2.72768	2.56811	3.39643	8.63109	0.78280	
	*r_j_*	0.16509	0.19456	0.41458	0.00402	0.09005	
WSO	*x_j_*	0.85459	5.26886	1.27613	6.15639	5.14336	6.644 × 10^−4^
	*y_j_*	2.81100	2.12723	3.72400	7.77702	0.99677	
	*r_j_*	0.20048	0.06864	0.47351	0.00116	0.08057	
STOA	*x_j_*	1.01642	3.68255	2.93057	2.85610	5.88747	6.704 × 10^−3^
	*y_j_*	3.31514	0.14579	4.50559	6.96590	1.15675	
	*r_j_*	0.06486	0.18653	0.00451	0.01124	0.00424	

**Table 4 biomimetics-09-00134-t004:** Coordinates of the two adjacent DWB curves in Example 2.

Curves	Variables	Values				
		*i* = 0	*i* = 1	*i* = 2	*i* = 3	*i* = 4
(W)1(t)	** *p* ** _1,*i*_	(0.3, 0.3)	(0.8, 1.6)	(1.6, 1.8)	(2.5, 1.7)	/
	*r* _1,*i*_	0.13	0.15	0.19	0.17	/
${\left(W\right)}_{2}(t)$	** *p* ** _2,*i*_	(2.5, 1.7)	(3.4, 1.8)	(3.8, 1.3)	(3.4, 0.7)	(2.2, 0.3)
	*r* _2,*i*_	0.17	0.12	0.13	0.1	0.16

**Table 5 biomimetics-09-00134-t005:** Optimization results of merging into 5-degree DWB curves without endpoint preservation.

Methods	Variables	Optimal Results						Errors
		*j* = 0	*j* = 1	*j* = 2	*j* = 3	*j* = 4	*j* = 5	
BEESO	*x_j_*	4.14407	−13.63569	49.35647	−27.28675	27.29917	13.75634	7.936 × 10^−2^
	*y_j_*	11.01504	2.49772	−28.09682	50.41259	15.34664	6.33120	
	*r_j_*	0.31182	0.25410	0.58469	0.07399	0.00208	0.38995	
HHO	*x_j_*	1.95800	3.58062	32.85162	−33.19215	34.00852	13.21403	1.896 × 10^0^
	*y_j_*	13.22078	−32.08853	48.85257	−15.32603	36.52032	6.02862	
	*r_j_*	0.28222	0.49638	1.09575	0.09179	0.67192	0.03881	
GWO	*x_j_*	2.96220	−0.14555	24.40576	−8.19088	20.28811	14.18703	6.706 × 10^−1^
	*y_j_*	12.35794	−16.21483	16.68298	−0.38029	39.69358	4.48880	
	*r_j_*	0.02199	0.17890	1.87352	0.09694	0.44597	0.01420	
CSA	*x_j_*	2.86664	−2.35508	34.38615	−25.52135	30.70782	13.21770	9.830× 10^−1^
	*y_j_*	13.43952	−27.95697	35.67889	−10.03823	41.66521	4.46278	
	*r_j_*	0.22263	0.93622	0.14821	0.32336	0.58220	0.12940	
SO	*x_j_*	3.95336	−10.73586	42.78311	−21.33005	25.50833	13.72622	1.259 × 10^−1^
	*y_j_*	11.51607	−2.07501	−19.33357	41.54719	19.30209	6.09531	
	*r_j_*	0.01003	0.37421	1.91634	0.00117	0.00131	0.33275	
COOT	*x_j_*	2.78067	3.58024	13.05992	6.89678	12.87292	14.74629	2.991 × 10^−1^
	*y_j_*	11.71382	−4.26839	−16.28066	41.79826	18.29245	6.19255	
	*r_j_*	0.28702	0.34258	0.67924	0.33386	0.05145	0.35011	
WSO	*x_j_*	4.20564	−13.24443	46.90530	−23.03463	24.78209	14.00221	1.081 × 10^−1^
	*y_j_*	11.29270	−1.16410	−19.59455	40.83427	20.10784	5.89122	
	*r_j_*	0.31442	0.01275	0.71721	0.08365	0.66488	0.15256	
STOA	*x_j_*	3.83290	−11.98145	46.50427	−23.51064	24.94221	13.94746	7.952 × 10^−1^
	*y_j_*	12.24440	−15.57719	18.53611	−8.39220	45.53216	4.00000	
	*r_j_*	0.01679	0.89726	1.59033	0.00101	0.91595	0.42818	

**Table 6 biomimetics-09-00134-t006:** Optimization results of merging into 5-degree DWB curves in the endpoint-preserving case.

Methods	Variables	Optimal Results				Errors
		*j* = 1	*j* = 2	*j* = 3	*j* = 4	
BEESO	*x_j_*	−8.20131	39.56450	−19.65931	24.34362	9.035 × 10^−2^
	*y_j_*	3.24733	−29.29145	49.90985	16.90465	
	*r_j_*	0.53727	0.00100	0.00100	0.56893	
HHO	*x_j_*	0.87732	13.14280	4.75472	16.71168	5.343 × 10^−1^
	*y_j_*	−3.53821	−7.58933	25.52112	25.73455	
	*r_j_*	0.31537	0.00101	0.84676	0.31721	
GWO	*x_j_*	0.47294	16.11563	2.13234	17.18804	3.160 × 10^−1^
	*y_j_*	2.97535	−28.24119	48.46377	17.45618	
	*r_j_*	0.02652	0.24920	0.17385	0.00476	
CSA	*x_j_*	−4.68901	29.32213	−9.49996	20.92611	1.310 × 10^−1^
	*y_j_*	3.30309	−29.24612	49.42552	17.16853	
	*r_j_*	0.39449	0.22791	0.02520	0.56299	
SO	*x_j_*	−7.39375	37.00421	−16.71285	23.27150	9.260 × 10^−2^
	*y_j_*	2.65403	−27.59176	48.18303	17.49296	
	*r_j_*	0.42663	0.16535	0.00100	0.53631	
COOT	*x_j_*	−8.55553	40.40914	−20.04365	24.39021	1.656 × 10^−1^
	*y_j_*	−1.89241	−14.78308	36.14274	21.37543	
	*r_j_*	0.01005	1.27617	0.47971	0.41079	
WSO	*x_j_*	−8.35262	40.14078	−20.35664	24.58088	9.188 × 10^−2^
	*y_j_*	2.59905	−27.38672	48.11605	17.48977	
	*r_j_*	0.13278	0.65601	0.43303	0.06355	
STOA	*x_j_*	0.40560	16.82436	0.36679	17.99242	3.627 × 10^−1^
	*y_j_*	−1.91517	−15.74742	38.24499	20.50645	
	*r_j_*	0.18871	0.00133	0.04571	0.00142	

**Table 7 biomimetics-09-00134-t007:** Coordinates of the two adjacent DWB curves in Example 3.

Curves	Variables	Values				
		*i* = 0	*i* = 1	*i* = 2	*i* = 3	*i* = 4
${\left(W\right)}_{1}(t)$	** *p* ** _1,*i*_	(2, 7.2)	(−1, 1)	(2.5, 0.1)	(5, 1.5)	/
	*r* _1,*i*_	0.1	0.3	0.4	0.2	/
${\left(W\right)}_{2}(t)$	** *p* ** _2,*i*_	(5, 1.5)	(7.5, 3)	(7.3, 7.8)	(3, 7)	(2.5, 2.5)
	*r* _2,*i*_	0.2	0.4	0.3	0.2	0.2

**Table 8 biomimetics-09-00134-t008:** Optimization results of merging into 4-degree DWB curves without endpoint preservation.

Methods	Variables	Optimal Results					Errors
		*j* = 0	*j* = 1	*j* = 2	*j* = 3	*j* = 4	
BEESO	*x_j_*	2.33358	−7.16306	11.94368	11.57288	1.92318	8.957 × 10^−2^
	*y_j_*	6.87698	0.49849	−13.06687	18.74353	2.21263	
	*r_j_*	0.08517	0.70571	0.12370	0.46669	0.18610	
HHO	*x_j_*	0.41095	7.43913	−0.72978	21.80887	1.33647	1.242 × 10^0^
	*y_j_*	6.81754	−4.44108	−2.78939	4.60870	4.11749	
	*r_j_*	0.00100	0.90287	0.00100	1.21484	0.14700	
GWO	*x_j_*	1.23982	0.87643	6.23859	16.36787	1.47955	2.491 × 10^−1^
	*y_j_*	7.17040	−1.71070	−11.53857	17.46736	2.32996	
	*r_j_*	0.00320	0.20739	0.14215	0.83420	0.05256	
CSA	*x_j_*	1.83074	−2.98093	8.27760	15.37562	1.49060	1.754 × 10^−1^
	*y_j_*	7.34904	−3.57450	−9.40825	14.95596	2.64042	
	*r_j_*	0.00995	0.85501	0.22883	0.54432	0.15462	
SO	*x_j_*	2.23359	−6.68287	11.81961	11.63043	1.85129	1.082 × 10^−1^
	*y_j_*	7.17724	−1.95063	−10.98384	16.47034	2.50440	
	*r_j_*	0.16209	0.15410	0.45796	0.27153	0.21107	
COOT	*x_j_*	1.88280	−3.70309	8.92993	14.75841	1.56662	1.214 × 10^−1^
	*y_j_*	6.90014	0.24457	−12.88773	18.62515	2.23042	
	*r_j_*	0.20332	0.03849	0.51155	0.21145	0.19090	
WSO	*x_j_*	2.38130	−6.98332	11.44939	12.27388	1.81599	9.278 × 10^−2^
	*y_j_*	6.80409	0.83965	−13.17581	18.78726	2.21304	
	*r_j_*	0.11398	0.21875	0.54759	0.09084	0.21260	
STOA	*x_j_*	1.59611	−2.29075	9.36872	13.10693	1.81932	1.821 × 10^−1^
	*y_j_*	6.76207	1.85716	−14.39966	20.78648	1.78510	
	*r_j_*	0.00199	1.04260	0.02212	0.00139	0.12128	

**Table 9 biomimetics-09-00134-t009:** Optimization results of merging into 4-degree DWB curves in the endpoint-preserving case.

Methods	Variables	Optimal Results			Errors
		*j* = 1	*j* = 2	*j* = 3	
BEESO	*x_j_*	−6.39142	12.63184	9.15882	1.467 × 10^−1^
	*y_j_*	−1.40489	−11.68342	16.97125	
	*r_j_*	0.61966	0.18928	0.38233	
HHO	*x_j_*	−6.96989	13.26127	8.72426	1.588 × 10^−1^
	*y_j_*	−1.97471	−10.73305	16.21491	
	*r_j_*	0.00164	0.64359	0.35071	
GWO	*x_j_*	−6.41252	12.65043	9.15650	1.574 × 10^−1^
	*y_j_*	−1.52899	−11.55796	16.89344	
	*r_j_*	0.03853	0.08682	0.67139	
CSA	*x_j_*	−6.41280	12.65905	9.13950	1.310 × 10^−1^
	*y_j_*	−1.42554	−11.65713	16.95251	
	*r_j_*	0.67248	0.11643	0.43568	
SO	*x_j_*	−6.39179	12.63240	9.15842	1.467 × 10^−1^
	*y_j_*	−1.40506	−11.68327	16.97116	
	*r_j_*	0.61948	0.18948	0.38217	
COOT	*x_j_*	−6.40350	12.64754	9.14612	1.468 × 10^1^
	*y_j_*	−1.42201	−11.66099	16.95623	
	*r_j_*	0.54193	0.28413	0.31917	
WSO	*x_j_*	−6.38955	12.62835	9.16185	1.467 × 10^−1^
	*y_j_*	−1.40627	−11.68131	16.96950	
	*r_j_*	0.62091	0.18820	0.38252	
STOA	*x_j_*	−6.34216	12.53565	9.25998	1.618 × 10^−1^
	*y_j_*	−1.69692	−11.50584	16.83058	
	*r_j_*	0.00544	0.00163	0.80658	

## Data Availability

All data generated or analyzed during the study are included in this published article.

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
