# Peer review of "Ameliorated Snake Optimizer-Based Approximate Merging of Disk Wang–Ball Curves"

_biomimetics, 2024, doi:10.3390/biomimetics9030134_

Round 1

Reviewer 1 Report

Comments and Suggestions for Authors

The authors propose a modified snake optimizer method for disk Wang-Ball curve merging, aiming to minimize merging errors. They claim that the model is designed to be more accurate and speedy, and its effectiveness in solving approximate merging models is demonstrated through numerical examples. They also mentioned that the method offers a new approach for computer-aided geometric design.

However, my comments should be addressed before the manuscript is considered for publication. 

1.      The authors suggested updating the keywords by selecting more relevant terms. Keywords play an important role in the appearance of the manuscript in a scholar's search, which will give it more hits and citations. 

2.      The authors should clearly justify the need for their approach in the presence of many effective approaches and state the pros and cons of their suggested method. 

3.      The results were presented in plots and simply re-stated in the text, but there is no in-depth discussion. 

4.      Combining optimization methods with cryptographic schemes is a widely used approach. Here are some recent related references you should consider to update the introduction and the related work sections: 

- https://doi.org/10.1007/s11227-020-03144-x

5.      The conclusion should be abstracted and self-contained to reflect the outcomes achieved by the research. Therefore, it should be rewritten. 

6.      In its current state, the level of English throughout the manuscript needs improvement. You may wish to ask a native speaker to check your manuscript for grammar, style, and syntax. 

Comments on the Quality of English Language

Would like to review the revised version of the manuscript.

Reviewer 2 Report

Comments and Suggestions for Authors

Overall, this manuscript presents valuable work that contributes to the field of bio-inspired algorithms for computational geometry. The structure and logic are complete. However, there are some details needs to be further clarified. 

Please see this list of comments regarding the issues needs to be clarified.

Line 46. Need explaination on what is CAD/CAM when it first appears.

Line 49. Need explaination on what are interval curve and disk curve. What is the main difference?

Line 78. "how to extenng and thus implemented a merging algorithm" not sure what it means. Please rewrite this sentence for better clarity.

Line 84-85. could be better written as "to investigate an approximation methogy to merge disk Wang-Ball curves with improved robustness and accuracy."

Line 86. How is the problem of merging curves becoming an optimization problem needs to be clarified.

Line 170. Define Min.

Line 172 and 177. Please use the proper form to define a minimization function. I see we are minimizing the error term and you need to make it clear that is achieved by selecting proper t. And t should be within what range?

Line 199. LB stands for lower bound, and UB stands for upper bound.

Line 209. What are the numerical type and range of k and K?

Line 218,219. "r" was used to represent the random number. Please use the same notation consistently. Is the "rand" here the same or different? Please claify or modify.There are many uses of the same instance in later context. Please correct all of them. Further, in line 259 there is a mixed use of "r" and "rand" so they perhaps are the same? Please unify the defination and use.

Line 224. There seems to be no fitness function defined in the context.Please define your fitness function.

Line 283-284. Please refer to Eq. 2.9 and Eq. 2.10 as the objective functiosn here for better clarity. Also, please explain why the two models, endpoint preserving and non-endpoint preserving, are discussed.

Line 298-299. Please explain why setting 0.6 as the threshold.

Figure 1. What are k and K initiated as?

Figure 2. The reviewer recommends not using such thick curves and big dots to represent the curves and the control points.

Figure 3 and 4. a-h. The color of curves overlapped due to thick curves. Please update the graphical representations to better show the differences between the merged curves. What does the variation in curve thickness in different sections represent? What does the color of those curve represent?

Comments on the Quality of English Language

Although there are many grammatic issues in this work, the English is readable and understandable. Please go through a through editing process to make the language reads smoother.

The reviewer here lists out some obvious points of improvement, but please perform an editing in more depth.

Line 11. Why "basis of the modified snake optimizer" has the abbrieviation as BEESO? 

Line 12-13 "difficult merging" to "difficulties in the merging".

Line 43. replace "reason" with "cause".

Line 44. add a comma after "Interval analysis"

Line 58. Abbreviations only needs to be spelled out once. Please check the entire manuscript to correct this.

Line 103. "informed" to "inferred"
